# *BcSOC1* Promotes Bolting and Stem Elongation in Flowering Chinese Cabbage

**DOI:** 10.3390/ijms23073459

**Published:** 2022-03-22

**Authors:** Yudan Wang, Xiu Huang, Xinmin Huang, Wei Su, Yanwei Hao, Houcheng Liu, Riyuan Chen, Shiwei Song

**Affiliations:** College of Horticulture, South China Agricultural University, Guangzhou 510642, China; ydwang@stu.scau.edu.cn (Y.W.); huangxiu18@126.com (X.H.); myshelley@126.com (X.H.); susan_l@scau.edu.cn (W.S.); yanweihao@scau.edu.cn (Y.H.); liuhch@scau.edu.cn (H.L.)

**Keywords:** flowering Chinese cabbage, *BcSOC1*, bolting, stem elongation, cell expansion, floral transition

## Abstract

Flowering Chinese cabbage is one of the most economically important stalk vegetables. However, the molecular mechanisms underlying bolting, which is directly related to stalk quality and yield, in this species remain unknown. Previously, we examined five key stem development stages in flowering Chinese cabbage. Here, we identified a gene, *BcSOC1* (*SUPPRESSOR OF OVEREXPRESSION OF CONSTANS1*), in flowering Chinese cabbage using transcriptome analysis, whose expression was positively correlated with bolting. Exogenous gibberellin (GA_3_) and low-temperature treatments significantly upregulated *BcSOC1* and promoted early bolting and flowering. Additionally, *BcSOC1* overexpression accelerated early flowering and stem elongation in both *Arabidopsis* and flowering Chinese cabbage, whereas its knockdown dramatically delayed bolting and flowering and inhibited stem elongation in the latter; the inhibition of stem elongation was more notable than delayed flowering. *BcSOC1* overexpression also induced cell expansion by upregulating genes encoding cell wall structural proteins, such as *BcEXPA11* (cell wall structural proteins and enzymes) and *BcXTH3* (xyloglucan endotransglycosidase/hydrolase), upon exogenous GA_3_ and low-temperature treatments. Moreover, the length of pith cells was correlated with stem height, and BcSOC1 interacted with BcAGL6 (AGAMOUS-LIKE 6) and BcAGL24 (AGAMOUS-LIKE 24). Thus, *BcSOC1* plays a vital role in bolting and stem elongation of flowering Chinese cabbage and may play a novel role in regulating stalk development, apart from the conserved function of *Arabidopsis* *SOC1* in flowering alone.

## 1. Introduction

*Brassica* spp. (family Cruciferae) are cultivated worldwide because of their economically important traits, including flavor [1,2], leafy heads, and enlarged organs (roots, stems, and inflorescences) [3,4]. Although several studies have focused on the development of leafy heads, roots, and stem tubers, there have been only a few studies on stem development in *Brassica* spp. [5,6]. Flowering Chinese cabbage (*Brassica campestris* L. ssp. *chinensis* (L.) Makino var. *utilis* Tsenet Lee), a subspecies of Chinese cabbage from southern China, is a typical stalk vegetable [7]. It has 10 chromosomes, and its genome is of the AA type. The genome size of *B. rapa*, a representative species of the AA genome, is 353.14 Mb [8]. As a diploid, flowering Chinese cabbage is suitable for gene modification and genetic studies. The edible parts of flowering Chinese cabbage, which include flower stalks and leaves, are rich in anticarcinogenic compounds and antioxidants such as glucosinolates, phenols, vitamin C, and carotenoids [1,7,9]. Development of the stem (also referred to as stalk) includes bolting (stem elongation and thickening) and flowering, both of which are directly related to stalk quality and yield. However, the molecular mechanisms underlying stalk development in flowering Chinese cabbage remain to be elucidated. The primary pathways regulating bolting and flowering in this species may be autonomous or dependent on FRIGIDA (FRI) [10].

The genetic pathways underlying floral transition that are responsive to endogenous and environmental stimuli have been elucidated [11]. In *Arabidopsis*, six important regulatory pathways, namely autonomous, photoperiod, age, vernalization, ambient temperature, and gibberellin (GA) pathways, govern flowering [12,13]. The primary integrators of multiple flowering pathways are *FLOWERING LOCUS T* (*FT*), *SUPPRESSOR OF OVEREXPRESSION OF CONSTANS1* (*SOC1*), and *LEAFY* (*LFY*) [14]. *SOC1*, which belongs to the MADS-box transcription factor family, regulates different stages of plant development, such as meristem differentiation, flowering time, and floral organ development [11,15]. In *Arabidopsis*, SOC1 interacts with AGAMOUS-LIKE 24 (AGL24) in the shoot apical meristem to directly regulate flowering factors *LFY* and *APETALA1* (*AP1*) and to promote flowering [16,17]. FLOWERING LOCUS C (FLC) interacts with SHORT VEGETATIVE PHASE (SVP) to inhibit *SOC1* transcription by directly interacting with the *SOC1* promoter in the vernalization pathway [18,19]. Gibberellin (GA_3_) regulates *LFY* transcription in the shoot apical meristem independent of or dependent on the *SOC1* pathway, thereby regulating floral transition [20].

Homologs of *SOC1* with conserved functions in flowering have been identified and characterized in numerous plant species. However, *SOC1* may also perform varied functions across different species. For example, the overexpression of *UNSHAVEN* (*UNS*), a gene isolated from *Petunia hybrida* and with the highest similarity to *SOC1*, resulted in early flowering and transformation of petals into leaf-like organs [21]. Strawberry *FvSOC1* regulates flowering in the shoot apical meristem and the differentiation of lateral buds into stolons or lateral rosettes through GA_3_ biosynthesis [22]. Furthermore, *SOC1*-like genes may alter the duration of dormancy instead of floral transition in kiwifruit [23].

Our previous study indicated that GA_3_ and low-temperature treatments promoted early bolting and flowering in flowering Chinese cabbage [24]. However, the specific molecular mechanism by which GA_3_ and low temperature regulate bolting in flowering Chinese cabbage are still rarely studied. In the present study, using RNA-seq data [25], we screened a bolting-related gene *BcSOC1*. *SOC1* is an important flowering regulator in plants that integrates multiple pathways including GA and ambient temperature pathways to regulate bolting and flowering. Therefore, we analyzed the expression of *BcSOC1* in the GA and low temperature response pathway for vernalization to characterize the function of *BcSOC1* and explore the molecular mechanisms of *BcSOC1* that regulate the bolting of flowering Chinese cabbage. The findings of this study will provide insights into the molecular mechanisms underlying stalk development in flowering Chinese cabbage and other stalk vegetables.

## 2. Results

### 2.1. Expression and Subcellular Localization of BcSOC1 in Flowering Chinese Cabbage

In the previous study, we examined five developmental stages (S1–S5) of the flowering Chinese cabbage cultivar ‘Youlv 501 caixin’ to gain insight into stalk development. With this aim, we performed RNA-seq using two biological replicates of shoot tip specimens from S1 (15 d after sowing, the seedling stage), S3 (31 d after sowing, the bolting stage), and S5 (37 d after sowing, the flowering or harvesting stage) [25]. We further analyzed differentially expressed flowering-related genes at these developmental stages (Appendix A). Interestingly, *BcSOC1* was significantly upregulated at stages S3 and S5, indicating that *BcSOC1* was a potential regulator of bolting and flowering in this species. In addition, the flowering-related genes *AP1*, *LFY*, *SPL3*, *SPL5*, and *SPL9* exhibited higher expression levels at stages S3 or S5 (Appendix A). Given the specificity of *Bc**SOC1* expression, we aimed to determine its role in bolting and flowering of flowering Chinese cabbage.

Based on the reference transcriptome sequences of flowering Chinese cabbage, two genes, namely *BcSOC1-1* and *BcSOC1-2,* were cloned and analyzed. Motif and domain analyses revealed that *BcSOC1* belonged to the MADS-box family of transcription factors, with highly conserved MADS-box, K-box domain, and *SOC1* motif (Figure 1A). Furthermore, sequence analyses indicated that *BcSOC1* was closely related to *SOC1* in other crucifer species; *BcSOC1-1* exhibited 100% homology with *BrSOC1* and *BnSOC1*, whereas *BcSOC1-2* exhibited 100% homology with *BjSOC1*. Moreover, *BcSOC1-1* and *BcSOC1-2* shared 93% and 94% homology, respectively, with *AtSOC1* (Figure 1A,B).

To further elucidate the role of *BcSOC1*, we examined the expression profile of *BcSOC1* in the roots, shoot apex, leaves, and flowers at the blooming stage of flowering Chinese cabbage using qRT-PCR (Figure 1C). *BcSOC1-1* transcript levels were significantly higher in the shoot apex than in the other organs, whereas *BcSOC1-2* was upregulated in the leaves and roots. Subcellular localization of *BcSOC1* revealed that both *BcSOC1-1* and *BcSOC1-2* were localized in the nucleus (Figure 1D and Appendix A).

### 2.2. GA_3_ and Low-Temperature Treatments Affect Bolting and BcSOC1 Expression in Flowering Chinese Cabbage

Previous studies have shown that both GA_3_ and low-temperature treatments promote bolting and flowering in flowering Chinese cabbage [24]. Therefore, to determine whether *BcSOC1* is involved in the regulation of bolting and flowering, the seeds of flowering Chinese cabbage were treated with GA_3_ (200 mg/L) and low temperatures (4 °C or 15 °C), and then cultured in perlite. As shown in Appendix A, plant height slowly increased during the early stages and rapidly after bolting. Furthermore, GA_3_ treatment promoted bolting, as indicated by increased height, during the entire growth period. The stem diameter of the control (CK) plants increased throughout the growth period, whereas GA_3_ treatment significantly reduced stem diameter. In the CK plants, budding and flowering time were 25 d and 30 d after sowing, respectively, whereas the budding time and flowering time were advanced to 21 d and 26 d after sowing, respectively, upon GA_3_ treatment. Additionally, budding and flowering rates were higher in the GA_3_-treated plants than in the CK plants.

*BcSOC1-1* expression was slightly affected by the GA_3_ treatment during the vegetative phase; however, GA_3_ notably affected the expression of *BcSOC1-1* during the reproductive phase (Figure 2A,B). *BcSOC1-1* expression remained low 8–18 d after the GA_3_ treatment and was not significantly different from that in the CK plants. However, 21–31 d after the GA_3_ treatment, the expression level of *BcSOC1-1* in the leaves rapidly increased 2.82–7.26 times and in the shoot tips 2.16–2.67 times compared with that in the CK plants. Furthermore, the expression level of *BcSOC1-2* in the leaves was upregulated 7.00 and 47.95 times compared with that in the CK leaves after 21 d and 31 d of treatment, respectively (Figure 2C). However, the expression level of *BcSOC1-2* in the shoot tips was upregulated 18.0, 35.82, 2.82, and 2.09 times compared to the CK plants at 13, 18, 25, and 31 d after treatment, respectively (Figure 2D). Therefore, the expression of *BcSOC1-2* was more significantly affected by GA_3_ treatment than that of *BcSOC1-1*.

Similarly, low temperatures promoted bolting in flowering Chinese cabbage, thereby increasing plant height and reducing stalk diameter (Appendix A). The expression levels of *BcSOC1-1* and *BcSOC1-2* in the leaves were significantly higher at low temperatures than in the CK leaves at 20, 22, and 25 d after treatment (Figure 2E,G). Similarly, *BcSOC1-1* and *BcSOC1-2* were significantly upregulated in the shoot tip at each sampling period, especially 20 d after treatment, which indicated the critical period of bolting and stem elongation in flowering Chinese cabbage (Figure 2F,H). In addition, the upregulation of *BcSOC1-2* expression in shoot tips was much higher than that of *BcSOC1-1* at 20 d after the low-temperature treatment, indicating that *BcSOC1-2* expression was more significantly affected by low temperature compared with that of *BcSOC1-1*. These results indicate that *BcSOC1* may be involved in the regulation of bolting, flowering, and stem elongation in flowering Chinese cabbage, as its expression is affected by both GA_3_ and low temperature.

### 2.3. BcSOC1 Overexpression Accelerates Stem Elongation and Flowering in Arabidopsis and Flowering Chinese Cabbage

To investigate whether *BcSOC1* plays the same role as *At**SOC1* in *Arabidopsis*, we ectopically expressed *BcSOC1-1* and *BcSOC1-2* under the constitutive CaMV 35S promoter in *Arabidopsis* accession Col. We obtained 25 pBI121:*BcSOC1-1* and 30 pBI121:*BcSOC1-2* T1 transgenic lines. The bolting and flowering time of all the transgenic lines with the WT background was decreased, with different severity of the phenotype (Figure 3A, Appendix A). The pBI121:*BcSOC1-1* transgenic plants (OE-1, 2, and 3) flowered 3–4 d earlier than the WT plants, and the phenotype of pBI121:*BcSOC1-2* transgenic plants (OE-4 and 5) was similar to that of pBI121:*BcSOC1-1* plants. However, a considerably early flowering phenotype was also observed in pBI121:*BcSOC1-2* transgenic plants, with the OE-6 line exhibiting the strongest phenotype (flowering 8 d earlier than the WT) (Table 1 and Appendix A). Transgenic lines were also significantly taller than the WT, with OE-6 line being the tallest (Figure 3A, Table 1). Interestingly, qRT-PCR analyses revealed that the earliest flowering line (OE-6) exhibited the highest level of transgene expression, and the severity of the phenotype positively correlated with the expression of *BcSOC1-1* and *BcSOC1-2* (Figure 3B,C). These results suggest that *BcSOC1* plays a role similar to that of *AtSOC1*, which promotes bolting and flowering in *Arabidopsis*. In addition, pBI121:*BcSOC1-2* lines had earlier bolting and flowering times than pBI121:*BcSOC1-1* did (Table 1). Compared with WT, the expression levels of pBI121:*BcSOC1-1* and pBI121:*BcSOC1-2* were increased 12–20 and 30–44 times, respectively. This suggests that the overexpression level of *BcSOC1-2* was higher than that of *BcSOC1-1*.

To genetically characterize *BcSOC1*, we optimized *Agrobacterium*-mediated transformation in flowering Chinese cabbage (Appendix A) and expressed *BcSOC1* under the CaMV 35S promoter. All six pBI121:*BcSOC1* T1 transgenic lines exhibited early bolting, accelerated stem elongation before flowering, and a significantly taller phenotype compared with the WT. Moreover, the first flower buds appeared markedly earlier in the transgenic lines compared with those in the WT plants, and pBI121:*BcSOC1-2* transgenic lines exhibited a more significant phenotype compared to the pBI121:*BcSOC1-1* lines (Figure 3D, Table 1). Consistent with these results, *BcSOC1* and its downstream gene *BcLFY* were significantly upregulated in the transgenic lines (Figure 3E–G), indicating that *BcSOC1* overexpression promoted early bolting, flowering, and stem elongation in flowering Chinese cabbage.

### 2.4. BcSOC1 Silencing Delays Bolting and Stem Elongation in Flowering Chinese Cabbage

We generated *BcSOC1* knockdown lines using VIGS (Figure 4) and obtained nine single and six double knockdown lines. We observed and analyzed the phenotypes of control (WT and pTRV2) and knockdown plants when control plants began to flower. Compared with the WT and empty vector control, the knockdown lines exhibited delayed bolting and flowering, increased leaf area and leaf number, and prolonged vegetative growth (Figure 4A–D and Appendix A). Among them, the phenotype of pTRV2:*BcSOC1-2* knockdown lines was more prominent than that of pTRV2:*BcSOC1-1* lines, and the phenotype of the double knockdown lines was more notable than that of the single knockdown lines. The expression of *BcSOC1* in the knockdown lines was significantly reduced compared with that in the CK plants, and the transgene expression was more significantly decreased in the double knockdown lines than in the single knockdown lines (Figure 4E–G). These results demonstrate that reduced *BcSOC1* expression in the knockdown lines resulted in delayed bolting in flowering Chinese cabbage, and the functions of the two homologous genes were overlapping and complementary.

We also examined the expression of *BcLFY*, which acts downstream of *Bc**SOC1*, in the knockdown lines [17] and observed that *BcLFY* was significantly downregulated in the knockdown lines compared with that in the CK plants. Furthermore, the downregulation was more remarkable in the double knockdown lines (Figure 4H).

We selected three knockdown lines (T2, T5, and T7) for further characterization. Notably, as the CK plants entered the end of flowering, the knockdown lines were just blooming and were significantly shorter than the CK plants. This indicates that stem elongation was inhibited before flowering in the knockdown lines (Figure 4I,J) and stems reached optimal height only at the final flowering stage (Figure 4J). These results demonstrate that *BcSOC1* knockdown not only delayed bolting and flowering but also inhibited stem elongation to some extent. Additionally, stem elongation inhibition was more significant than delayed flowering, and the phenotype was most significant in T7, followed by T5, which suggested that *BcSOC1-2* had the more effective silencing compared with *BcSOC1-1*. These findings are consistent with the results of overexpression of *BcSOC1-2* in *Arabidopsis* and flowering Chinese cabbage.

### 2.5. BcSOC1 Overexpression Probably Induces Cell Expansion by Upregulating BcEXPA11 and BcXTH3

To determine whether cell size was affected in the transgenic plants, we compared the size of pith cells in longitudinal sections of the shoot apex at the blooming stage of both WT and overexpression lines (Figure 5A). Compared to the WT, cell length and area in the pBI121:*BcSOC1-1* and pBI121:*BcSOC1-2* transgenic plants were significantly increased (Figure 5B,C). In a previous study, RNA-seq of flowering Chinese cabbage exposed to 15 °C for 5 d [26] revealed a significant upregulation of cell wall structural proteins and enzymes, including *BcEXPA11* and xyloglucan endotransglycosidase/hydrolase *BcXTH3*, at low temperature (Appendix A). Therefore, we determined the expression of *BcEXPA11* and *BcXTH3* in the shoot apex of pBI121:*SOC1* and pTRV2:*BcSOC1* lines. Both genes were upregulated in *BcSOC1* overexpression lines OE-3 and OE-6 and downregulated in the knockdown lines T2, T5, and T7 (Figure 5D,E). These results suggest that the increase in stalk length was positively correlated with increased pith cell size, which was promoted by the overexpression of *BcSOC1* through a significant upregulation of the *BcEXPA11* and *BcXTH3* expression.

We also examined the expression of *BcEXPA11* and *BcXTH3* after GA_3_ and low-temperature (15 °C) treatments (Figure 5F,G). In the CK group, both genes were significantly upregulated in the shoot apex after 25 or 30 d of sowing. However, GA_3_ and low-temperature treatments significantly upregulated *BcEXPA11* and *BcXTH3*, with the expression levels rapidly increasing after 25 or 30 d of sowing, which is the critical period for stem elongation in flowering Chinese cabbage.

### 2.6. BcSOC1 Interacts with BcAGL6, BcAGL24, and BcSOC1

AtSOC1 forms dimers with AtAGL6 and AtAGL24 to regulate bolting and flowering in *Arabidopsis* [27]. To determine the putative protein interactions of *BcSOC1* in flowering Chinese cabbage, yeast two-hybrid experiments were performed using 53-BD+AD-T and LAM-BD+AD-T as positive and negative controls, respectively. We observed that BcSOC1-2 interacted with BcSOC1-1 and BcSOC1-2 to form homologous dimers and with BcAGL6 and BcAGL24 to form heterologous dimers (Figure 6A). However, BcSOC1-1 did not interact with BcAGL6 and BcAGL24 (Figure 6A).

To verify the results of the yeast two-hybrid experiments, bimolecular fluorescence complementation experiments were performed in vivo. Yellow fluorescent protein (YFP) fluorescence in the nuclei of plant cells confirmed the interactions of BcSOC1-2 with BcSOC1-1, BcSOC1-2, BcAGL6, and BcAGL24 (Figure 6B and Appendix A).

## 3. Discussion

### 3.1. SOC1 Expression Pattern Correlates with Bolting in Flowering Chinese Cabbage

Gene expression patterns are closely related to gene function. Previous studies have shown that *SOC1* expression in leaves promotes flowering, and it is necessary for the initiation of flowering in the meristem [28,29,30]. In this study, *BcSOC1-1* and *BcSOC1-2* were expressed in the same tissues but at different levels. In flowering Chinese cabbage, *BcSOC1-1* was upregulated in the shoot apex (Figure 1D), similar to *Chrysanthemum* [31], whereas *BcSOC1-2* was upregulated in the leaves (Figure 1D), similar to *Arabidopsis*, rice, and wheat [29,32,33]. Furthermore, *BcSOC1* was primarily expressed in vegetative organs, unlike *Fa**SOC1* in strawberry, which is upregulated in the reproductive organs, such as the shoot apex, flower bud, stamen, and sepal [34]. Previous studies on *Arabidopsis*, strawberry, and soybean suggest that SOC1-like proteins are localized in the cytoplasm [17,34,35]. However, in flowering Chinese cabbage, *BcSOC1* was localized in the nucleus (Figure 2E,F), similar to *ZmMAD1* and *LsSOC1* [36], indicating that *SOC1* plays different roles in different species.

*Cl**SOC1*-1 and *Cl**SOC1*-2 are induced throughout the growth period in *Chrysanthemum* [31]. The transcript levels of *SOC1* in rice and wheat are significantly increased during transition from vegetative to reproductive growth, and then maintained at high levels [32,33]. In the present study, the transcript levels of *BcSOC1-1* and *BcSOC1-2* were high throughout growth and development of flowering Chinese cabbage, and were significantly increased during floral transition (Appendix A). This expression profile is consistent with previous reports, indicating that *BcSOC1-1* and *BcSOC1-2* may be involved in the regulation of bolting and flowering in flowering Chinese cabbage.

### 3.2. GA_3_ and Low-Temperature Treatments Accelerate Bolting and Flowering in Flowering Chinese Cabbage by Mediating BcSOC1 Expression

*SOC1* responds to flowering signals from both endogenous and environmental stimuli and acts on the downstream floral meristem-specific genes to participate in the formation of plant floral meristems and the regulation of flowering time [12,15,37]. Moon et al. [20] reported that exogenous GA_3_ promoted *SOC1* expression in WT *A. thaliana* (L*er*) and shortened flowering time. GA_3_ treatment increased *SOC1* expression and rescued the non-flowering phenotype of *ga1-3*, whereas *SOC1* null mutant exhibited reduced sensitivity to GA_3_ for flowering. In this study, *BcSOC1-1* and *BcSOC1-2* were upregulated in flowering Chinese cabbage upon exposure to exogenous GA_3_ after budding (Figure 2A–D), which is consistent with the findings of Dorca-Fornell et al. [38] and suggests that *BcSOC1-1* and *BcSOC1-2* may mediate premature flowering caused by GA_3_. Moreover, rapid stem elongation and significant increase in plant height (Appendix A) indicate that *BcSOC1* regulates stem elongation in flowering Chinese cabbage.

Lee et al. [39] reported that *SOC1* did not express in *FRI FLC* lines of *Arabidopsis* without vernalization, whereas *SOC1* expression significantly increased after vernalization. In addition, *SOC1* expression increased upon vernalization treatment in *FLC-3* mutants [20]. In the present study, the expression profiles of *BcSOC1-1* and *BcSOC1-2* at 4 °C or 15 °C were the same as that under the GA_3_ treatment and were rapidly upregulated after budding (Figure 2E–H and Appendix A). This suggests that *BcSOC1-1* and *BcSOC1-2* regulate bolting in flowering Chinese cabbage when induced at low temperatures.

### 3.3. BcSOC1 Plays a Vital Role in Regulating Bolting and Stem Elongation in Flowering Chinese Cabbage

*At**SOC1* integrates different signaling pathways to promote bolting and flowering in *Arabidopsis* [28,39]. In the present study, *BcSOC1* overexpression in *Arabidopsis* (Col) shortened its flowering time (Figure 3, Table 1), suggesting that *BcSOC1* exhibits a high degree of functional conservation in promoting flowering. Additionally, increased plant height in *BcSOC1* overexpression lines indicated its role in stalk development. To test this hypothesis, we generated and phenotyped *BcSOC1* overexpression lines in the background of flowering Chinese cabbage, and we observed that *BcSOC1* overexpression significantly advanced bolting in flowering Chinese cabbage (Figure 3, Table 1). Furthermore, pTRV2:*BcSOC1* knockdown lines exhibited delayed bolting and inhibition of stem elongation before flowering (Figure 4), suggesting that *BcSOC1* regulates both bolting and stem elongation in flowering Chinese cabbage. Similar findings were reported by Mauren et al. [40], who found that *MtSOC1a* promotes flowering and primary stem elongation in *Medicago*, further verifying the role of *SOC1* in the regulation of stalk development. In addition, we determined that the two homologous genes were functionally redundant, and *BcSOC1-2* was the key regulatory factor.

Several MADS-box transcription factors regulate the expression of downstream target genes in the form of dimers [12,15,37]. In *Arabidopsis*, the MADS-box transcription factors AGL6, AGL24, FUL, and SEP3 bind with the regulatory sequences of *SOC1* [27,41]. In the present study, BcAGL6, BcAGL24, BcSOC1-1, and BcSOC1-2 formed dimers with BcSOC1-2 (Figure 6). These data suggest that BcSOC1-2 may upregulate the expression of downstream meristem genes through interactions with other MDS-box proteins and thereby regulate bolting of flowering Chinese cabbage. However, BcSOC1-1 did not interact with BcAGL6, BcAGL24, and BcSOC1-1 (Figure 6A). Nayar et al. [42] conducted deletion analysis and revealed that the KC region (K-box and C-terminal domain) plays a pivotal role in dimerization. However, K-box and C-terminal domains are not completely conserved relative to the highly conserved MADS-box. Alternative splicing events were found only at the borders of or within the K-box domain; a small change in the sequence of K-box domain transcript can have strong consequences for the interaction capacity and functioning of the encoded protein [43]. In the present study, amino acid mutations were detected between the K-box of BcSOC1-1 and BcSOC1-2, which may be the main reason for the distinct interaction patterns between the two BcSOC1 proteins with interacting partners.

Cell cycle regulation of the shoot apical meristem affects stem elongation and thickening, and *BcEXPA11* and *BcXTH3* are the key factors responsible for cell expansion [44,45]. *EXPA11* promotes the sliding of polymers such as cell wall cellulose and hemicellulose by reducing the viscosity of polysaccharides between cell walls and enabling cell wall expansion [46]. *XTH3* participates in the regulation of cell wall relaxation by cutting xyloglucan chains [47]. Investigation of the pith cell structure in *Bc**SOC1* overexpression lines revealed that stem elongation was correlated with the increase in pith cell length and area (Figure 5). Therefore, the rapid elongation of stalk in flowering Chinese cabbage may be attributed to increased cell expansion. The present study revealed that GA_3_ and low temperatures accelerate bolting and flowering in flowering Chinese cabbage, probably by upregulating *BcSOC1*, *BcEXPA11*, and *BcXTH3* (Figure 2 and Figure 5), and the overexpression of *BcSOC1* induces cell expansion by upregulating *BcEXPA11* and *BcXTH3* (Figure 5). Figure 7 demonstrates the role of *BcSOC1* in the bolting and flowering of flowering Chinese cabbage.

While numerous *SOC1*-like genes regulate flowering, others may exhibit both functional conservation and divergence with *AtSOC1* [22,23]. Taken together, the results of the present study suggest that *BcSOC1* may function as a floral integrator gene and promote stem elongation, providing new insights into stalk development in flowering Chinese cabbage. In addition, stalk development in flowering Chinese cabbage is accompanied by both vegetative and reproductive growth, which is different from other cruciferous leaf vegetables such as *B. rapa*, *B. oleracea*, and *B. campestris*. Therefore, further investigation is necessary to understand the synergistic mechanisms underlying vegetative and reproductive growth in regulating stalk development.

## 4. Materials and Methods

### 4.1. Plant Materials and Growth Conditions

Flowering Chinese cabbage cultivar ‘Youlv 501 caixin’ was obtained from the Guangzhou Academy of Agricultural Science and cultivated in a greenhouse at the College of Horticulture, South China Agricultural University. The wildtype (WT) and *BcSOC1* transgenic plants were grown at 25/20 °C (day/night) under 16/8 h light/dark cycles and light intensity of 100 μmol m^−2^ s^−1^. Knockdown lines generated by VIGS were cultured at 22/20 °C (day/night), and other environmental conditions were the same as above.

Seeds of *Arabidopsis thaliana* (L.) Heynh. accession Columbia (Col) were obtained from The *Arabidopsis* Information Resource (http://www.Arabidopsis.org/index.jsp, accessed on 19 March 2022), and all *Arabidopsis* plants were grown in a growth chamber at 22 °C under a 16/8 h light/dark cycle.

### 4.2. Gene Cloning

Total RNA was extracted from the leaves and shoot apex of flowering Chinese cabbage using a HiPure Plant RNA Mini Kit (Magen, Guangzhou, China) according to the manufacturer’s instructions. First-strand cDNA was synthesized and subjected to inverse transcription using a HiScript QRT SuperMix for qPCR Reagent Kit (with gDNA Wiper; Vazyme, Nanjing, China). Coding sequences (CDS) of *BcSOC1-1*, *BcSOC1-2*, and *BcLFY* isolated from the shoot apex were cloned using gene-specific primers (Appendix A).

### 4.3. Phylogenetic Analyses of SOC1 Homologs

Amino acid sequences of BrSOC1, BnSOC1, BjSOC1, BoSOC1, CfSOC1, AtSOC1, CcSOC1, GmSOC1, VvSOC1, MdSOC1, FvSOC1, PaSOC1, DeSOC1, TaAGL20, OsAGL20, ZmSOC1, and HaSOC1 were obtained from the National Centre for Biotechnology Information website (http://www.ncbi.nlm.nih.gov/, accessed on 19 March 2022) (Appendix A). Nucleotide and amino acid sequences were aligned using ClustalW and MUSCLE, respectively, and phylogenetic trees were constructed using the neighbor joining method. All steps of the phylogenetic reconstruction were performed in MEGA7 [48]. Support values were obtained using 1000 bootstrap replicates.

### 4.4. Quantitative Real-Time Polymerase Chain Reaction (qRT-PCR)

Total RNA was isolated from the leaves and stems of plants exposed to exogenous GA_3_ and low-temperature treatments. RNA of the WT and *BcSOC1* transgenic plants was isolated from the fourth leaf, and qRT-PCR was performed using a ChamQ SYBR Color qPCR Master Mix (Vazyme) on a LightCycler 480 real-time PCR instrument (Roche Diagnostics, Rotkreutz, Switzerland). Three biological and three technical replicates were used for each sample, and glyceraldehyde-3-phosphate dehydrogenase (*GAPDH*) was used as the internal reference to normalize gene expression. The primers used in qRT-PCR analyses are listed in Appendix A.

### 4.5. Subcellular Localization

Full-length *BcSOC1-1* and *BcSOC1-2* CDS were cloned into pEAQ-EGFP vectors and fused with green fluorescent protein (GFP) under the control of the CaMV 35S promoter. The primers used for cloning are listed in Appendix A.

*Nicotiana benthamiana* leaves were first infiltrated with *Agrobacterium tumefaciens* strain GV3101 transformed with corresponding constructs and nuclear localization signal (NLS-DsRed). After three days, GFP fluorescence and DsRed protein were detected at 448 nm and 550 nm, respectively, using a laser-scanning confocal microscope. To further verify the results of subcellular localization, the plasmids were transformed into onion epidermal cells using a previously described method [49].

### 4.6. GA_3_ and Low-Temperature Treatments

Seeds of flowering Chinese cabbage were sterilized using 70% ethanol for 3 min, washed 3–4 times with deionized water, and then placed in a petri dish lined with filter paper. Thereafter, 200 mg/L GA_3_ solution and the same amount of deionized water were added to GA_3_ and control (CK) treatments, respectively. Another set of sterilized seeds was exposed to low temperatures in a growth chamber, one group was placed at 4 °C and a second group at 15 °C. After five days of low-temperature treatment, the seeds were sown in perlite. Seedlings with three leaves and one core were transplanted into hydroponic containers and supplied with half-strength Hoagland nutrient solution.

The plants were harvested at 37–40 days after sowing. Plant height (cm) and stem thickness (mm) were measured using the cross method with ruler and vernier caliper, respectively. For GA_3_ treatment, plant height and stem diameter were measured at 19, 21, 23, 25, and 27 d, bud rate was calculated at 21, 23, 25, 27, 29, 31, 33, and 35 d, and flowering rate was calculated at 26, 28, 30, 32, 34, 36, 38, and 40 d after the treatment. For low-temperature treatment, plant height and stem diameter were measured at 17, 19, 21, 23, and 25 d, bud rate was calculated at 20, 22, 24, 26, 28, 30, 32, 34, and 36 d, and flowering rate was calculated at 23, 25, 27, 29, 31, 33, 35, 37, and 39 d after the treatment. Leaves and shoot apex were collected at 8, 10, 13, 16, 18, 21, 25, 31 d and 13, 16, 18, 21, 25, 31 d, respectively, after GA_3_ treatments, and at 9, 12, 15, 17, 20, 22, 25 d and 12, 15, 17, 20, 22, 25 d, respectively, after low-temperature treatments. Samples were rapidly frozen in liquid nitrogen and stored at −80 °C for RNA extraction (three biological replicates per treatment, with 10 plants per replicate).

### 4.7. Ectopic Expression of BcSOC1 in Arabidopsis

The pBI121:*BcSOC1* construct, generated by cloning *BcSOC1* CDS into pBI121-GFP under the control of CaMV 35S promoter, was first transformed into *A. tumefaciens* strain GV3101, which was then transformed into *Arabidopsis* plants using the floral-dip method [50]. Transformed lines were selected on Murashige and Skoog (MS) medium (pH 5.8) containing 1% sucrose, 0.7% agar, and 50 mg/L kanamycin (Kan). The primer sequences are listed in Appendix A.

### 4.8. Virus-Induced Gene Silencing (VIGS) Assay

*BcSOC1-1* and *BcSOC1-2* CDS were cloned into pTRV2 plasmids to generate silencing vectors pTRV2:*BcSOC1-1* and pTRV2:*BcSOC1-2*, respectively, and VIGS experiments were performed as previously described [51]. The pTRV2:*BcSOC1* and pTVR1 vectors were co-infiltrated into the cotyledons of young seedlings of the flowering Chinese cabbage. Two weeks after infection, the second true leaf was collected from each plant, and RNA was extracted for qRT-PCR analysis to evaluate the efficiency of gene silencing. Plants co-infiltrated with pTRV2 and pTRV1 were used as negative controls. Subsequently, plant morphological characteristics were determined. The primers used for VIGS assay are listed in Appendix A.

### 4.9. Agrobacterium-Mediated Transformation in Flowering Chinese Cabbage

Cotyledon transformation of the ‘Youlv 501 caixin’ cultivar was performed using a previously described protocol with modifications [52]. After sowing on half-strength MS medium (pH 5.8) containing 1% sucrose and 0.6% agar, the sterilized seeds were placed in a tissue culture at 25/20 °C (day/night) under 16/8 h light/dark cycles and light intensity of 100 μmol m^−2^ s^−1^. The cotyledons and hypocotyls of sterile seedlings were excised 3 d after emergence. The cotyledon and hypocotyl explants were then pre-cultured for callus initiation on MS medium (pH 5.8) containing 0.3 mg/L 1-naphthaleneacetic acid (NAA), 5.0 mg/L 6-benzylaminopurine (6-BA), 2.0 mg/L AgNO_3_, 3% sucrose, and 1% agar, in the dark for two days at 25 °C before inoculation and co-culturing with *A. tumefaciens* EHA105. The preincubated hypocotyls were soaked in *Agrobacterium*-infection buffer (MS medium containing 3% sucrose and 1 mg/L AS [pH 5.8]) for 10 min and transferred to the co-cultivation medium (MS medium containing 0.3 mg/L NAA, 5.0 mg/L 6-BA, 2.0 mg/L AgNO_3_, 3% sucrose, and 1% agar [pH 5.8]) in the dark for two days at 25 °C. After one week of delayed selection (MS medium containing 0.3 mg/L NAA, 5.0 mg/L 6-BA, 2.0 mg/L AgNO_3_, 3% sucrose, 1% agar, 200 mg/L timentin (Tim) [pH 5.8]), the explants were transferred to callus- and shoot-induction media (MS medium containing 0.3 mg/L NAA, 5.0 mg/L 6-BA, 2.0 mg/L AgNO_3_, 3% sucrose, 1% agar, 200 mg/L Tim, and 15 mg/L Kan [pH 5.8]). The secondary medium was changed every two weeks. When the regenerating shoots reached a height of 2–3 cm, they were transferred to the rooting medium (MS medium containing 0.3 mg/L NAA, 3% sucrose, 1% agar, and 200 mg/L Tim [pH 5.8]) to obtain Kan-resistant plantlets.

### 4.10. Yeast Two-Hybrid Assay

*BcSOC1-1*, *BcSOC1-2*, *BcAGL6*, and *BcAGL24* CDS were cloned into pGADT7 or pGBKT7 plasmids to generate yeast two-hybrid vectors. The recombinant vector was then delivered into yeast strain Y2HGold to generate fusion proteins according to a previously described protocol [53]. Primer information is given in Appendix A.

### 4.11. Bimolecular Fluorescence Complementation Assay

The vectors were produced by amplifying the CDS of *BcSOC1-1*, *BcSOC1-2*, *BcAGL6*, and *BcAGL24* without the stop codons from their corresponding cDNAs. The fragments were cloned into the N- or C-terminus of pSPYNE-35S and pSPYCE-35S vectors to generate fusion proteins [54]. The *Agrobacterium* strain harboring the recombination plasmid and DsRed was co-infiltrated into young tobacco leaves as previously described [53]. After two days of incubation, yellow fluorescent protein (YFP) fluorescence was detected at 448 nm using a laser-scanning confocal microscope. To indicate the nuclei, DsRed protein was detected at 550 nm. Primer sequences are listed in Appendix A.

## 5. Conclusions

In this study, we showed that GA_3_ and low-temperature treatments significantly upregulated *BcSOC1*, *BcEXPA11*, and *BcXTH3*, thereby promoting early bolting and flowering in flowering Chinese cabbage. Furthermore, *BcSOC1* knockdown dramatically delayed bolting, flowering, and inhibited stem elongation in this species. We also elucidated the molecular mechanism underlying stem elongation in flowering Chinese cabbage, wherein the overexpression of *BcSOC1* induced cell expansion by upregulating *BcEXPA11* and *BcXTH3*. We further demonstrated that BcAGL6 and BcAGL24 interacted with BcSOC1 to regulate bolting in flowering Chinese cabbage. These findings provide the foundation for further investigations on the mechanisms underlying stalk development in cruciferous stalk vegetables.

## Figures and Tables

**Figure 1 ijms-23-03459-f001:**
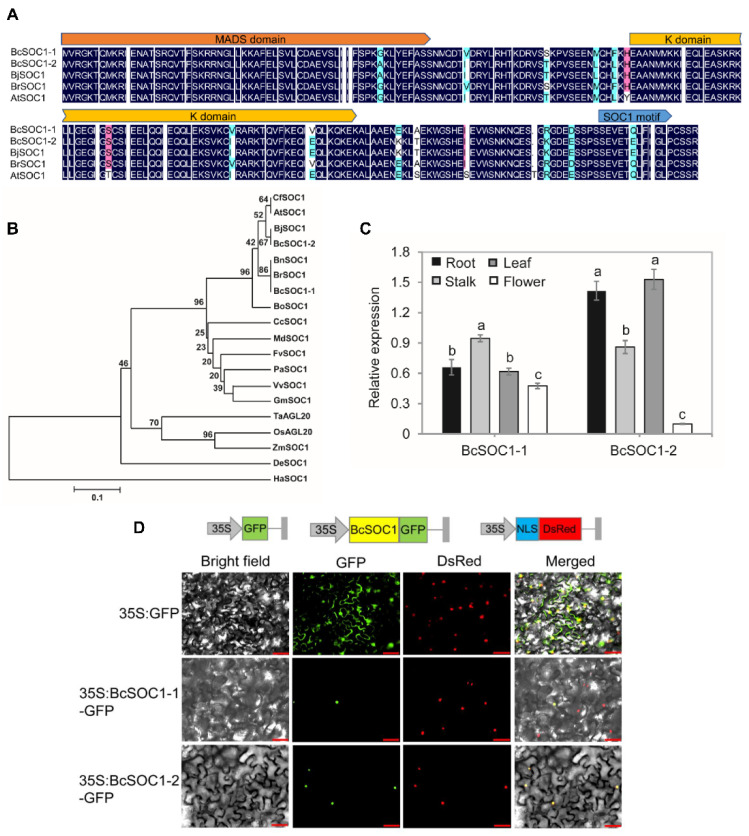
Sequence and expression analysis of *BcSOC1* in flowering Chinese cabbage. (**A**) Amino acid sequence analysis of BcSOC1 and other SOC1-like proteins. BcSOC1 was used as the query sequence, and other SOC1-like proteins with highly similar sequences in *Brassica juncea*, *B. rapa,* and *Arabidopsis thaliana* were retrieved from the NCBI database. (**B**) Phylogenetic analysis of SOC1 sequences of different plant species was performed using the neighbor joining method in MEGA7, with 1000 bootstrap replicates. Bo: *B. oleracea*, Bn: *B. napus*, Bc: *B. campestris*, Cf: *Cardamine flexuosa*, Md: *Malus domestica*, Fv: *Fragaria vesca*, Pa: *Prunus armeniaca*, De: *Dendrobium*, Ta: *Triticum aestivum*, Os: *Oryza sativa*, Zm: *Zea mays*, Ha: *Helianthus annuus*, Gm: *Glycine max*, Vv: *Vitis vinifera*, Cc: *Carya cathayensis*. (**C**) Expression level of *BcSOC1* in different tissues (a, b and c) of flowering Chinese cabbage. (**D**) Subcellular localization of *BcSOC1* in *Nicotiana benthamiana.* Bar = 50 μm.

**Figure 2 ijms-23-03459-f002:**
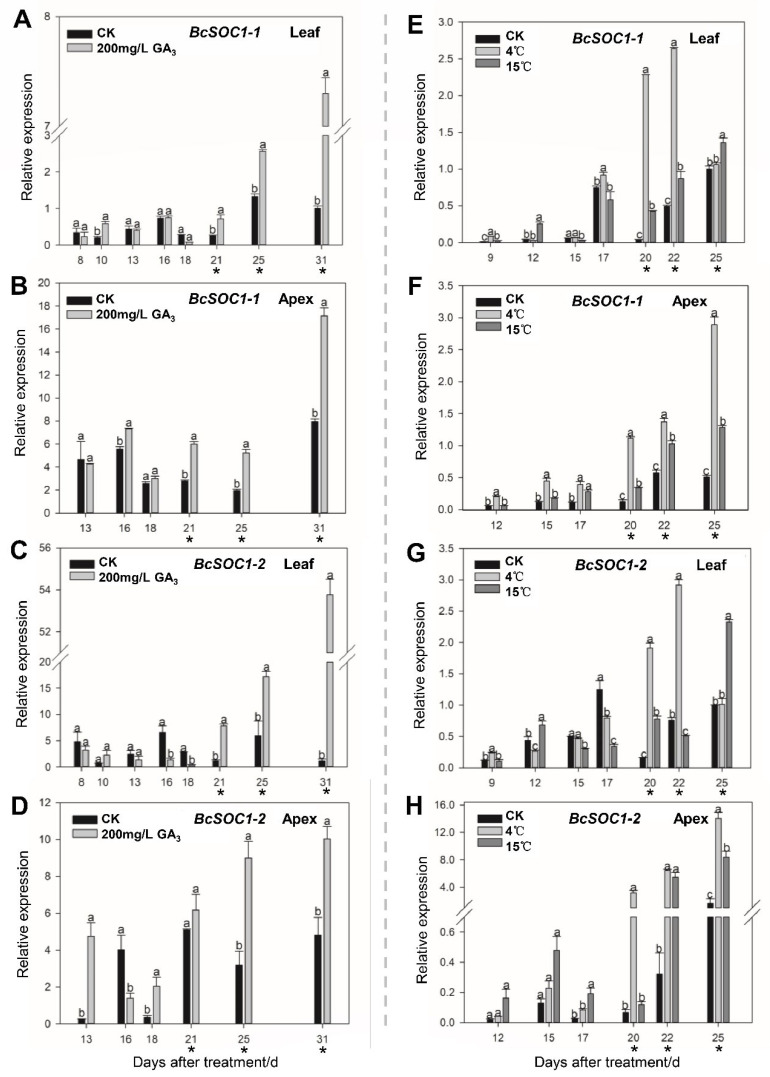
Effects of GA_3_ and low-temperature treatments on *BcSOC1* expression in flowering Chinese cabbage. (**A**–**D**) Expression levels of *BcSOC1-1* and *BcSOC1-2* in the leaves and shoot apex exposed to GA_3_. (**E**–**H**) Expression levels of *BcSOC1-1* and *BcSOC1-2* in the leaves and shoot apex exposed to low temperature (4 °C and 15 °C). CK as the control. Error bars represent standard errors. Different letters (a, b and c) indicate significant differences (*p* < 0.05) determined using ANOVA. * represents the reproductive phase.

**Figure 3 ijms-23-03459-f003:**
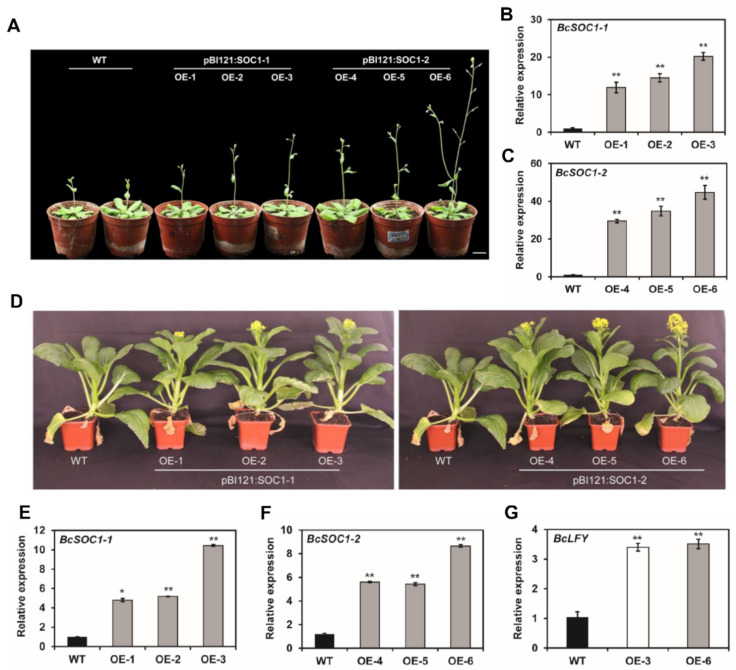
Overexpression of *BcSOC1* in *Arabidopsis* and flowering Chinese cabbage. (**A**) Ectopic expression of *BcSOC1* accelerated early flowering phenotype in *Arabidopsis*. Bar = 2 cm. (**B**,**C**) Expression analyses of *BcSOC1-1* and *BcSOC1-2* using qRT-PCR in wild type (WT) and overexpression lines. (**D**) Phenotypic characterization of the WT and pBI121:*BcSOC1* transgenic lines. Bar = 2 cm. (**E**–**G**) Expression of *BcSOC1-1*, *BcSOC1-2*, and *BcLFY* in the overexpression lines. WT as the control (CK). Significant differences were determined using Student’s *t*-test (* represents *p* < 0.05 and ** represents *p* < 0.01).

**Figure 4 ijms-23-03459-f004:**
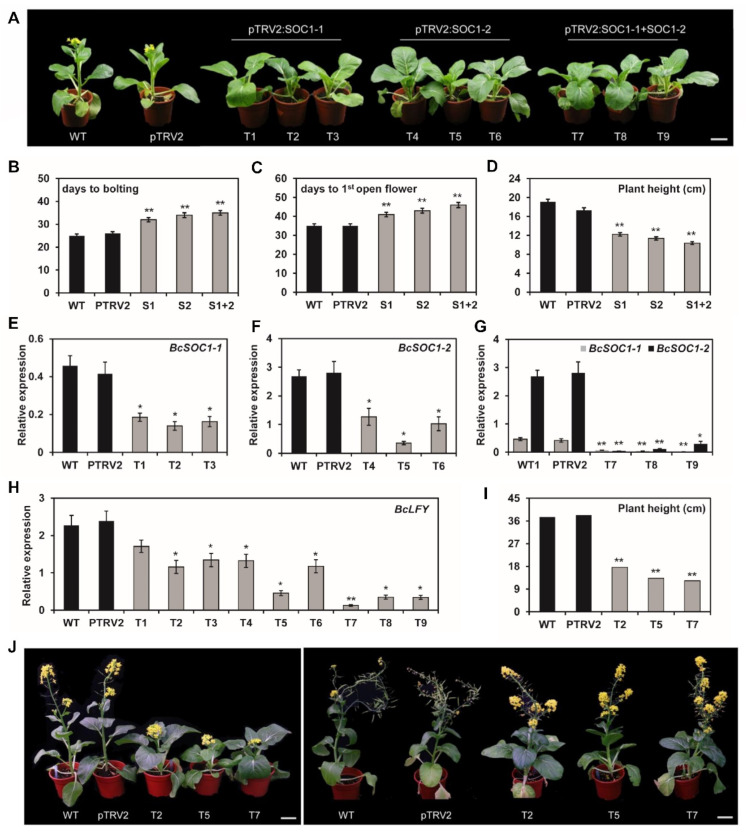
Knockdown of *BcSOC1* using virus-induced gene silencing (VIGS) in flowering Chinese cabbage. (**A**) Silencing of *BcSOC1* significantly delayed bolting in flowering Chinese cabbage. S1, S2, and S1+2 represent pTRV2:*SOC1-1*, *SOC1-2*, and *SOC1-1* + *SOC1-2*, respectively. (**B**–**D**) Quantification of delayed bolting phenotypes in *BcSOC1* knockdown lines. S1, S2 and S1+2 represent averages of the corresponding single and double knockdown lines, respectively. (**E**–**H**) Expression of *BcSOC1-1*, *BcSOC1-2*, and *BcLFY* in different *BcSOC1* knockdown lines. (**I**,**J**) Silencing of *BcSOC1* inhibited stem elongation in flowering Chinese cabbage, and the plant height was significantly decreased. WT and pTRV2 as the control (CK). Error bars represent standard errors. Significant differences were determined using Student’s *t*-test (* represents *p* < 0.05 and ** represents *p* < 0.01). Bar = 5 cm.

**Figure 5 ijms-23-03459-f005:**
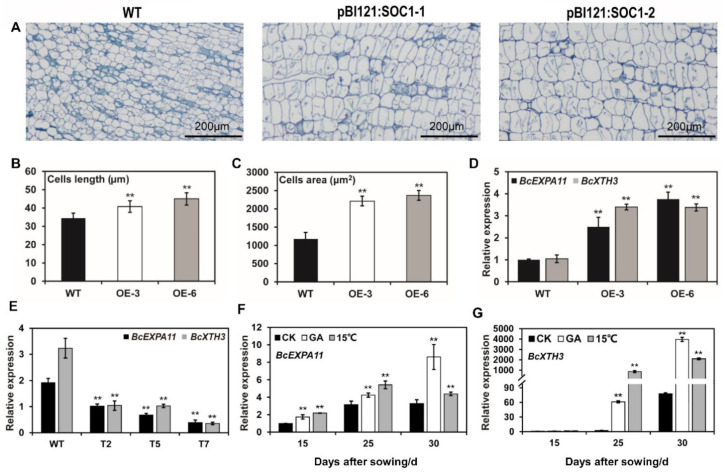
Expression of *BcEXPA11* and *BcXTH3* correlates with cell expansion. (**A**) Pith cell microstructures in the longitudinal sections of the shoot apex of wild type (WT) and *BcSOC1* transgenic lines. Bar = 200 μm. (**B**,**C**) Pith cell length and area in the shoot apex. (**D**,**E**) Relative expression of cell expansion-related genes *BcEXPA11* and *BcXTH3* in *BcSOC1* overexpression and knockdown lines. Error bars represent standard errors. Significant differences were determined using Student’s *t*-test (** represents *p* < 0.01). (**F**,**G**) Expression of *BcEXPA11* and *BcXTH3* under GA_3_ and low-temperature (15 °C) treatments. CK as the control. Significant differences were determined using ANOVA.

**Figure 6 ijms-23-03459-f006:**
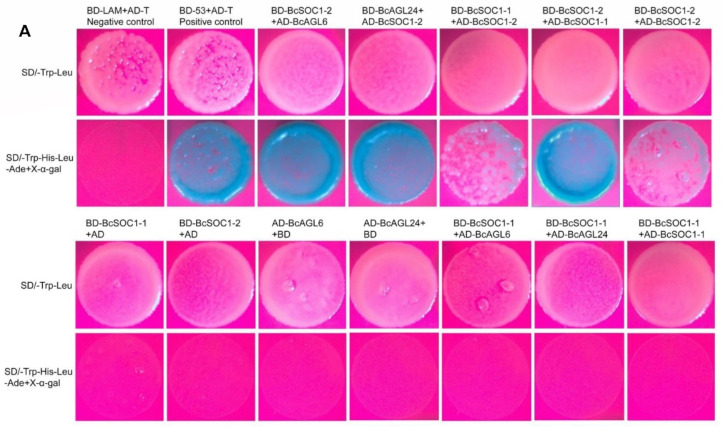
Yeast two-hybrid and bimolecular fluorescence complementation assays to determine protein interactions. (**A**) Yeast two-hybrid assay for protein–protein interactions between BcSOC1 and BcAGL6, BcAGL24, and BcSOC1. BD and AD represent empty pGBKT7 and pGADT7 vectors, respectively. SD/-Trp-Leu, the synthetic dextrose medium lacked tryptophan and leucine; the SD/-Trp-His-Leu-Ade medium lacked tryptophan, histidine, leucine, and adenine. Positive bacteria were stained using X-α-Gal. (**B**) Bimolecular fluorescence complementation assay was used to detect the interactions of BcSOC1 (fused with N-terminal fragment of YFP) with BcAGL6, BcAGL24, and BcSOC1-1 (fused with C-terminal fragment of YFP). The nuclei were stained with DsRed. Bar = 50 μm.

**Figure 7 ijms-23-03459-f007:**
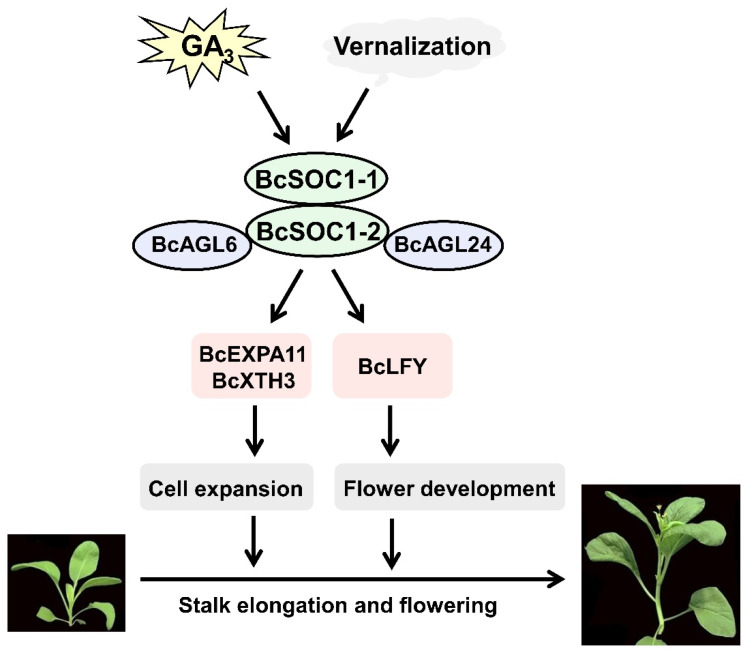
Model demonstrating GA_3_- and low temperature-induced bolting and stalk elongation in flowering Chinese cabbage. The arrow represents promotion.

**Table 1 ijms-23-03459-t001:** Quantification of early bolting and flowering phenotypes in *BcSOC1* transgenic lines.

Species	Lines	Days to Flowering (d)	Days to Bolting (d)	Plant Height (cm)
	WT	31.58 ± 0.96 a	/	5.11 ± 0.86 c
*Arabidopsis*	pBI121:*SOC1-1*	27.72 ± 0.74 b	/	9.83 ± 1.33 b
	pBI121:*SOC1-2*	25.77 ± 1.22 c	/	16.84 ± 3.96 a
	WT	38.92 ± 1.08 a	30.08 ± 0.90 a	19.03 ± 1.05 c
*B. campestris*	pBI121:*SOC1-1*	32.92 ± 1.51 b	24.92 ± 1.24 b	25.80 ± 1.21 b
	pBI121:*SOC1-2*	29.83 ± 1.47 c	23.17 ± 1.19 c	28.06 ± 1.09 a

Data are mean ± standard error (*n* = 12). Different letters (a, b and c) indicate significant differences (*p* < 0.05) determined using ANOVA.

## Data Availability

Not applicable.

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
