# Peer review of "BcSOC1 Promotes Bolting and Stem Elongation in Flowering Chinese Cabbage"

_ijms, 2022, doi:10.3390/ijms23073459_

Round 1

Reviewer 1 Report

In this manuscript the authors explore the biological role of flowering Chinese cabbage genes BcSOC1-1 and BcSOC1-2. They genetic and molecular approaches to show that BcSOC1-1 and BcSOC1-2 promotes bolting and stem elongation. They partially elucidate the molecular mechanism showing that BcSOC1-2 proteins interact with other MADS box factors like AGL6 and AGL24 as well as forming homodimers and that BcSOC1-1 and BcSOC1-2 regulate the cell wall structural proteins BcEXP11 and BcXTH3.

The manuscript is well presented and well writen

Some minor points

-line 90 ''..given the specificity of BcSOC1, we aimed to determine..." what specificity are the authors refer to? it is not clear

-fig1 need better quality photos, its not easy to see the results

-Fig1D the GFP signal in 35S:BcSOC1-2 does not seem to quite colocalize with dsRED, can you explain this or provide better photo?

-line 128, CK are the control plants?

-line 147 must mean Figure S2 not S1

-Figure 2 a and b that indicate statistical significance authors must present the meaning in the legend of the figure

-line 170 the data for the flower time and bolting of all the transgenic lines could be made available as supplementary

-lines 184-185 authors draw the conclusion that BcSOC1-2 may be more critical than BsSOC1-1, the genes are expressed at different levels so maybe this is the reason 1-2 look more prominent

-Fig3 flowering time (days to flowering, days to bolting and days to first open flower) data may be more appropriate to be presented as tables not as bar graphs since the differences are usually small and harder to see, also the number of plants used in flowering experiments is important and should be shown

-Fig 6 false colours on yeast two hybrid photos do not help present the data, the actual photos with the yeast cultures should presented

also S6 the rest of the Y2H data should be presented as regular Figure, they are important cause they differentiate the action of the two BcSOC1 genes

line 301''expressed in different tissues...'' they are not expressed in defferent tissues, they are expressed in the same tissues but at different levels

Reviewer 2 Report

This is a solid work focusing on the functional analysis of SOC1 in Chinese cabbage. The authors used a wide range of physiological and molecular tests to reveal the role of BcSOC1 in the regulation of flowering. The experiments are well planned and executed, appropriate statistical tests were used to support the conclusions. The key points of the results are that (i) BcSOC1 regulates flowering initiation in a very similar manner compared with its counterpart in Arabidopsis (AtSOC1), but (ii) in contrast to AtSOC1, BcSOC1 modulates cell expansion and thus stem elongation as well.

I have a few minor comments on the manuscript, the consideration of these could make the text even clearer and more precise.

  1. Please explain “CK” in all figure legends. I found the legend of Figure S5 and the section of M&M where CK is explained as control, but this should be shown in the legend of all figures.
  2. Line 147: Did you intend to refer to Figure S2 here?
  3. Please provide the time of day of harvesting in the M&M section.
  4. Line 184: “This suggests 184 that BcSOC1-2 may be more critical than BcSOC1-1.” I would not say this, because the stronger effect of BcSOC1-2 overexpression can be explained by the higher level of overexpression compared with the BcSOC1-1 lines.
  5. Figure 3. G,H,I: what do SOC1-1 and SOC1-2 mean? The averages of the 3 transgenic lines? Please explain in the legends. The same comes up in Figure 4. B,C,D : do S1, S2 and S1+2 represent averages of the corresponding knockdown lines?
  6. Line 231-232. “which 231 suggested that BcSOC1-2 may be the key gene responsible for the phenotype.” Again, I would not say this, since the stronger effect of SOC1-2 knockdowns can be explained by the more effective silencing compared with the WT plants.
  7. Line 244. “2.5. BcSOC1 Overexpression Induces Cell Expansion by Upregulating BcEXPA11 and BcXTH3”. This is a too strong and misleading subtitle. Cell expansion and upregulation of EXPA/XTH3 genes indeed show a correlation, but the available data are not sufficient to establish a cause-effect relationship between them (i.e. I cannot see the proof that cell expansion is induced by the upregulation of those genes). An acceptable change would be the insertion of ‘likely’ or ‘probably’ or something like these.
